# Identification of Key Genes and Biological Pathways Associated with Skeletal Muscle Maturation and Hypertrophy in *Bos taurus*, *Ovis aries,* and *Sus scrofa*

**DOI:** 10.3390/ani12243471

**Published:** 2022-12-08

**Authors:** Fatemeh Mohammadinejad, Mohammadreza Mohammadabadi, Zahra Roudbari, Tomasz Sadkowski

**Affiliations:** 1Department of Animal Science, Faculty of Agriculture, Shahid Bahonar University of Kerman, Kerman 7616914111, Iran; 2Department of Animal Science, Faculty of Agriculture, University of Jiroft, Jiroft 7867155311, Iran; 3Department of Physiological Sciences, Faculty of Veterinary Medicine, Warsaw University of Life Sciences, 02-776 Warsaw, Poland

**Keywords:** hypertrophy, gene expression, gene ontology, muscle development

## Abstract

**Simple Summary:**

One of the traits considered in livestock production is the gain of muscle mass, which is one of the factors responsible for governing the growth of skeletal muscles. Muscle hypertrophy involves the proliferation and differentiation of muscle cells and the maturation of muscle fibers. Advances in animal genetics and breeding rely on identifying hub genes, understanding ontological features, and identifying pathways of gene activity. Therefore, the aim of this study was to identify the hub genes and mechanisms involved in skeletal muscle maturation and hypertrophy in livestock species (*Bos taurus*, *Ovis aries*, and *Sus scrofa*). The hub genes identified in this study can be used to better identify and improve the growth and maturation of skeletal muscles and, as a result, enhance meat quality characteristics for breeding purposes in the mentioned species.

**Abstract:**

The aim of the current study was to identify the major genes and pathways involved in the process of hypertrophy and skeletal muscle maturation that is common for *Bos taurus*, *Ovis aries*, and *Sus scrofa* species. Gene expression profiles related to *Bos taurus*, *Ovis aries*, and *Sus scrofa* muscle, with accession numbers GSE44030, GSE23563, and GSE38518, respectively, were downloaded from the GEO database. Differentially expressed genes (DEGs) were screened out using the Limma package of R software. Genes with Fold Change > 2 and an adjusted *p*-value < 0.05 were identified as significantly different between two treatments in each species. Subsequently, gene ontology and pathway enrichment analyses were performed. Moreover, hub genes were detected by creating a protein–protein interaction network (PPI). The results of the analysis in *Bos taurus* showed that in the period of 280 dpc–3-months old, a total of 1839 genes showed a significant difference. In *Ovis aries*, however, during the period of 135dpc–2-months old, a total of 486 genes were significantly different. Additionally, in the 91 dpc–adult period, a total of 2949 genes were significantly different in *Sus scrofa*. The results of the KEGG pathway enrichment analysis and GO function annotation in each species separately revealed that in *Bos taurus*, DEGs were mainly enriched through skeletal muscle fiber development and skeletal muscle contraction, and the positive regulation of fibroblast proliferation, positive regulation of skeletal muscle fiber development, PPAR signaling pathway, and HIF-1 signaling pathway. In *Ovis aries*, DEGs were mainly enriched through regulating cell growth, skeletal muscle fiber development, the positive regulation of fibroblast proliferation, skeletal muscle cell differentiation, and the PI3K-Akt signaling, HIF-1 signaling, and Rap1 signaling pathways. In *Sus scrofa*, DEGs were mainly enriched through regulating striated muscle tissue development, the negative regulation of fibroblast proliferation and myoblast differentiation, and the HIF-1 signaling, AMPK signaling, and PI3K-Akt signaling pathways. Using a Venn diagram, 36 common DEGs were identified between *Bos taurus*, *Ovis aries*, and *Sus scrofa*. A biological pathways analysis of 36 common DEGs in *Bos taurus, Ovis aries*, and *Sus scrofa* allowed for the identification of common pathways/biological processes, such as myoblast differentiation, the regulation of muscle cell differentiation, and positive regulation of skeletal muscle fiber development, that orchestrated the development and maturation of skeletal muscle. As a result, hub genes were identified, including *PPARGC1A, MYOD1, EPAS1, IGF2, CXCR4*, and *APOA1*, in all examined species. This study provided a better understanding of the relationships between genes and their biological pathways in the skeletal muscle maturation process.

## 1. Introduction

Muscles make up meat, one of the most desirable animal products, and meat represents a major source of animal protein in the human diet (both skeletal muscle and visceral meat). Muscle meat is mainly composed of skeletal muscle fibers and various amounts of connective and adipose tissue, as well as small amounts of epithelial and nervous tissue, which play a role in its quantitative and qualitative properties [1]. The fundamental mechanism underlying livestock growth is the accumulation of muscle and adipose tissue [2]. Being the largest tissue in the body, skeletal muscle represents one of the most important complexes of biologically active tissues in mammals [3]. Elucidating the mechanism(s) of muscle growth and the intrinsic properties of the muscle may provide information that allows improvements in the quality of meat in production [4]. Therefore, it is necessary to know the molecular background that regulates meat growth and metabolism.

Skeletal muscle development is mainly regulated by the function of muscle precursor cells—myoblasts [5]—and is a complex process that can be divided into prenatal and postnatal stages. Livestock muscle is developed in two main stages: the prenatal phase—proliferation (hyperplasia—an increase in number of cells) as well as myotube formation, and the postnatal phase—muscle growth by hypertrophy (an increase in the size of muscle fibers) [6]. Skeletal muscle matures during late gestation at approximately 105 days in sheep [7], 210 days in cattle [8], and 114 days in pigs [4].

As mentioned above, muscle development is characterized by hyperplasia and hypertrophy. Hypertrophy is an increase in the size of skeletal muscle as a result of an increase in the diameter and length of already-formed fibers. Satellite cells proliferation (population of former myoblasts that did not fuse into myotubes) is the main cause of hyperplasia and skeletal muscle regeneration during postnatal life. Skeletal muscle stem cells reside in the spaces between muscle fibers’ sarcolemma and their basal lamina. The satellite cells are able to self-renew, proliferate, and fuse with existing muscle fibers as a prerequisite of muscle hypertrophy. All above-mentioned stages of myogenesis are key periods for skeletal muscle growth and development, and they directly affect muscle growth potential in meat production [9].

Analysis of transcriptomic data is a tool of great importance that allows identification of the key differentially expressed genes (DEGs) at each time point of the fetal and postnatal muscle development and to investigate the associated regulatory factors [3]. A complex gene network regulates the quality of meat and carcass [10]. Therefore, the identification of relationships between genes and carcass quality traits is critical to understanding animal development directly influencing meat composition. Additionally, identifying the underlying mechanisms and pathways determining the quantity and quality of muscle and marbling in carcasses is critical to better address demands of customers for prime quality meat products [1].

The aim of the current study was to identify the key genes and pathways involved in the maturation and hypertrophy of the skeletal muscle of *Bos taurus*, *Ovis aries*, and *Sus scrofa* species using comprehensive bioinformatics analysis.

## 2. Materials and Methods

### 2.1. Preparation of Data

The Gene Expression Omnibus (GEO) database was used as a source of expression profile collection. Three gene expression profiles, including GSE44030, GSE23563, and GSE38518, derived from *Bos taurus, Ovis aries*, and *Sus scrofa*, respectively, were downloaded from the GEO database. The dataset GSE44030 contained 20 biological samples of *Bos taurus* muscle with expression data from 60 days postconception (dpc) up to 3 months of age. The dataset GSE23563 contained 40 biological samples of *Ovis aries* skeletal muscle (70-, 85-, 100-, 120-, 135-days postconception, birth, and 1 month, 2 months of age). The dataset GSE38518 contained 24 biological samples of skeletal muscle from *Sus scrofa* at different developmental stages (35, 63, 91 dpc, and adult samples). Raw data of 280 dpc and 3 months in *Bos taurus*, 135 dpc and 2 months in *Ovis aries*, and 91 dpc and adult state in *Sus scrofa* were taken into consideration during a targeted analysis to investigate biomarkers and biological pathways of skeletal muscle maturation and hypertrophy stage (Table 1).

### 2.2. Differentially Expressed Genes

The interactive web tool GEO2R was applied to identify DEGs between the samples. (http://www.ncbi.nlm.nih.gov/geo/geo2r/) (accessed on 1 January 2013) [11]. GEO2R is an effective online tool mostly used to compare two sets of samples in most GEO series to identify genes with differential expression under the same experimental conditions. The Limma package was used for differential expression analysis in this web tool. Criteria for DEGs identification were Fold Change > 2 and adjusted *p*-value ≤ 0.05. Common genes between three species were identified using a Venn diagram.

### 2.3. Biological Processes and KEGG Pathways Analysis

An overrepresentation enrichment analysis (ORA) was performed using the Database for Annotation, Visualization, and Integrated Discovery (DAVID) version 2021 (https://david.ncifcrf.gov/) (accessed on 1 December 2021) [12]. The DAVID database was utilized to conduct gene ontology (GO) analysis on biological processes, molecular functions, and cellular components, as well as pathway enrichment analysis using the Kyoto Encyclopedia of Genes and Genomes (KEGG). Gene ontology terms and KEGG pathways of the co-expressed DEGs were identified using the above online tool with a false discovery rate (FDR) of 5% to gain a better understanding of the biological mechanisms involved in skeletal muscle maturation and hypertrophy.

### 2.4. Network Analysis

The Search Tool for the Retrieval of Interacting Genes (STRING) version 11.5 (https://string-db.org) (accessed on 12 August 2021) was used to explore protein–protein interaction (PPI) networks and potential DEG interactions [13]. The protein–protein interaction networks of DEGs were derived from validated experiments [14]. A combined interaction score of >0.4 was considered significant. The PPI networks were visualized using Cytoscape software (http://www.cytoscape.org) (accessed on 12 August 2021) [15]. The *p*-value ≤ 0.05 was considered to indicate a statistically significant difference. The topology scores of the nodes in the PPI network were estimated based on closeness centrality, betweenness centrality, and degree centrality.

## 3. Results

### 3.1. Analysis of Differential Gene Expression

The analysis revealed that a total of 1839 genes were significantly different between the 280 dpc and 3-month stages in *Bos taurus*, of which 1107 genes were upregulated and 732 genes were downregulated. In *Ovis aries*, a total of 486 DEGs were identified between the stage of 135 dpc and 2 months, of which 240 genes were upregulated and 246 genes were downregulated. In *Sus scrofa*, a total of 2949 genes were significantly different between the stage of 91 dpc and adult, of which 1383 genes were upregulated and 1566 genes were downregulated. Common genes between three species were identified using a Venn diagram (Figure 1). The names of these genes identified between the three species are shown in the Appendix A. Among them, 36 genes were common between *Bos taurus, Ovis aries*, and *Sus scrofa* muscle (Table 2).

### 3.2. Biological Processes and KEGG Pathways Analysis

DEGs were used as the inputs of the online software of our DAVID pathway analyses. DAVID is helpful in explaining the genome-scale datasets by translating the data collection into biological meanings. First, different expression genes for each animal were analyzed separately. The results of the GO analysis in *Bos taurus* revealed that DEGs were significantly enriched, with 68 significant biological processes (*p*-value ≤ 0.05). The most important biological processes in which DEGs were related to the maturation and hypertrophy of skeletal muscle development are listed in Table 3. Among the more significant biological processes, the processes connected with the myogenesis of *Bos taurus* were skeletal muscle fiber development and skeletal muscle contraction.

For *Ovis aries*, the results of our enrichment analysis indicated 85 significant biological processes (*p*-value ≤ 0.05) associated to DEGs. The most important processes related to the final steps of skeletal muscle development—maturation and fiber hypertrophy—are shown in Table 4. Among them, the highest significance was assigned to the regulation of cell proliferation and growth.

Genes with different expressions in *Sus scrofa* skeletal muscle were analyzed for functional enrichment. The results revealed that these genes were linked to 106 significant biological processes (*p*-value ≤ 0.05), of which the most important in light of muscle maturation and hypertrophy—meat quality traits—are presented in Table 5. In this analysis, the most important biological processes are linked with cell proliferation, muscle tissue maturation, and cell cycle regulation.

Then, among the genes with different expressions, 36 common genes in *Bos taurus, Ovis aries*, and *Sus scrofa* were identified. Further analysis revealed that these genes were linked to 325 biological processes (*p*-value ≤ 0.05). The most significant biological processes are listed in Table 6, along with the genes assigned to them. The most significant skeletal muscle maturation and hypertrophy processes are: adipose tissue development, myoblast differentiation, and the regulation of muscle cell differentiation.

The results of the KEGG pathway analysis in *Bos taurus* showed that DEGs were significantly enriched, with 22 pathways (*p*-value ≤ 0.05). The most significant pathways are listed in Table 7.

Genes with different expression in *Ovis aries* skeletal muscle were analyzed for the KEGG pathway. The most important pathways are presented in Table 8.

For *Sus scrofa*, the results of the pathway indicated 53 significant pathways (*p*-value ≤ 0.05) associated to DEGs. The most significant pathways are listed in Table 9.

Then, KEGG pathway analysis was performed between the three species. The list of the most significant pathways are listed in Table 10.

### 3.3. PPI Network Construction and Hub Gene Identification

The PPI network was created using the 36 common DEGs by String database and Cytoscape software. Not-connected and partially connected proteins were omitted (Figure 2). Six key hub genes, including *PPARGC1A, MYOD1, EPAS1, IGF2, CXCR4*, and *APOA1*, were reported in *Bos taurus, Ovis aries*, and *Sus scrofa* after the network was analyzed based on degree. Among them, the most important was the peroxisome proliferator-activated receptor gamma coactivator 1-alpha, which manifested the highest closeness centrality, betweenness centrality, and degree centrality (Table 11).

## 4. Discussion

Skeletal muscle development represents a complex process, which involves the alteration of the growth of muscle cells, as well as the differentiation and maturation of muscle fiber, including hypertrophy—an increase in the size of muscle fibers [16]. Muscle hyperplasia (an increase in number of cells) and hypertrophy (muscle fiber growth in terms of length and diameter) both depend on the proliferation of myoblasts [17]. The contribution of hyperplasia to muscle growth is limited to prenatal or a limited time after birth, while muscle growth depends more on hypertrophy [18]. The present study was conducted to identify the key genes and mechanisms involved in the maturation and hypertrophy of skeletal muscle common for livestock species, namely *Bos taurus, Ovis aries*, and *Sus scrofa*. The results of the analysis showed a total of 1839, 486, and 2949 differentially expressed genes in a comparison of pre- and postnatal muscle in *Bos taurus, Ovis aries*, and *Sus scrofa*, respectively (Appendix A). Using BP and pathway analysis identification can help in the understanding of the essential mechanisms involved in skeletal muscle development, especially muscle maturation. The results of the KEGG pathway enrichment analysis and GO function annotation in all examined species revealed that DEGs were mainly enriched in skeletal muscle fiber development, skeletal muscle contraction, the positive regulation of fibroblast proliferation and skeletal muscle fiber development processes, ribosome biogenesis in eukaryotes, and the PPAR and HIF-1 signaling pathways in *Bos taurus* (Table 3 and Table 7); the regulation of cell growth, skeletal muscle fiber development, the positive regulation of fibroblast proliferation and skeletal muscle cell differentiation processes and ECM receptor interactions, and the PI3K-Akt, HIF-1, Rap1 signaling pathways in *Ovis aries* (Table 4 and Table 8); and the regulation of striated muscle tissue development, the negative regulation of fibroblast proliferation and myoblast differentiation processes and insulin signaling, the HIF-1, FoxO, AMPK signaling pathways, and thyroid hormone signaling pathways in *Sus scrofa* (Table 5 and Table 9).

Common genes for the skeletal muscle development process were identified by differential expression between the three species. In this research, we identified 36 genes responsible for the maturation and hypertrophy of skeletal muscle in *Bos taurus, Ovis aries*, and *Sus scrofa* (Figure 1, Table 2). Finally, the results of the GO and KEGG analyses revealed that common genes were mainly enriched in the biological processes and signaling pathways, of which the most important are myoblast differentiation; the differentiation of muscle cells; the positive regulation of skeletal muscle fiber development processes and PPAR; insulin; the ECM–receptor interaction; and insulin resistance (Table 6 and Table 10).

In this investigation, we used some bioinformatics tools to identify genes among of the common genes that play a key role in skeletal muscle maturation. Using the CytoNCA plugin for network analysis led us to finding some hub genes with significant expression changes that can be identified as biomarkers. Network analysis showed that six genes (*PPARGC1A, MYOD1, EPAS1, IGF2, CXCR4*, and *APOA1*) have a high degree of centrality, and they are referred to as hub genes because they are crucial for development in each species (Figure 2 and Table 11).

### 4.1. Myogenesis-Related Processes

The involvement of DEGs in the aforementioned processes and pathways seems to be a prerequisite of proper myogenesis, finally leading to muscle fiber maturation and hypertrophy. The top three biological processes associated with skeletal muscle maturation and hypertrophy are shown in Table 6.

Myoblast differentiation, including its regulation, is governed by multiple factors that order this multistep process to withdraw from the cell cycle and induce myoblast differentiation and fusion into multinucleated myotubes, which finally gives rise to muscle fibers [19]. Myogenic regulatory factors, such as myogenic differentiation 1 (MYOD1), are restrictedly expressed prenatally in somite-derived myogenic progenitor cells and their derived myoblasts, and are important in postnatal satellite cells regulation [20]. The development of muscle fibers occurs when myotubes mature, extend, and merge with other myoblasts and satellite cells [21]. The regulation of this phenomenon is an ordered multistep process that leads to skeletal muscle fiber growth [19], which is crucial for further muscle fiber maturation and hypertrophy. In our research, common genes related to the myoblast differentiation process were identified and are shown in Table 6. Among them, *JAG1, EPAS1, MYOD1*, and *IGF2* are also mentioned as hub genes crucial for myogenesis that are common for all three examined species (Figure 2, Table 6 and Table 11).

During myogenesis, the adipose tissue development begins, followed by an accumulation of fat in myofibers and between muscle bundles, which increases with age [22] Myoblasts and intramuscular adipocytes are derived from mesenchymal stem cells, which are able to differentiate into myogenic and non-myogenic cell lines. In cattle, adipogenic progenitor cells begin to differentiate into preadipocytes at ~180 days of gestation [23]. It is also known that skeletal muscle satellite cells can acquire the features of adipocytes (dysdifferentiation), express adipocyte-specific genes, and accumulate lipids [24]. This suggests that muscle cells and adipocytes interplay during growth and that such early events influence skeletal muscle adipogenesis, influencing intramuscular fat content and muscle structure [25]. In the present analysis, two adipogenesis-related genes were found as common genes, namely *DGAT2* and *NAMPT* (Table 6). The common activity of all aforementioned genes in *Bos taurus, Ovis aries*, and *Sus scrofa* muscle may indicate their key involvement in its maturation, including myofibers maturation, hypertrophy, and the gradual increase in intramuscular and intermuscular fat.

### 4.2. Common KEGG Pathways

As mentioned above, four KEGG pathways are common for the skeletal muscle development of *Bos taurus, Ovis aries*, and *Sus scrofa*, with PPAR and insulin signaling and insulin resistance pathways shown as the most statistically significant. Peroxisome proliferator-activated receptors (PPARs) are a class of nuclear receptors that play important roles in development and energy metabolism, the regulation of satellite cell proliferation, skeletal muscle regeneration, and the increase in muscle fiber type (oxidative or glycolytic) [26]. The PPARs have major impacts on muscle homeostasis, with PPARγ directly implicated in lipid deposition in muscle [27]. PPARs can also affect insulin signaling by multiple mechanisms. The insulin signaling pathway is responsible for glucose and lipid homeostasis, as well as proliferation and differentiation and protein synthesis/degradation, thus regulating muscle growth and hypertrophy. It is well known that insulin has an anabolic effect on growing skeletal muscle [28,29,30,31]. Moreover, PPAR-γ can also enhance insulin resistance by decreasing the production of proinflammatory mediators [32]. Numerous studies have confirmed that insulin resistance accompanied by mitochondrial dysfunction might suppress protein synthesis to induce the loss of skeletal muscle mass, and that it is also involved in regulationing myogenic differentiation [33].

In our analysis, the PPAR signaling pathway was represented by three common genes, namely ME1, APOA1, and PLIN2 (Table 10). The insulin signaling pathway and insulin resistance were also the pathways indicated as common for *Bos taurus, Ovis aries*, and *Sus scrofa*, with the activity of PYGM, PHKB, and PPARGC1A genes similar for all species (Table 10). The interplay of the above-mentioned genes belonging to both pathways seems to be vital for the muscle development of *Bos taurus, Ovis aries*, and *Sus scrofa*.

### 4.3. Hub Genes

The network analysis performed on the batch of genes common for three livestock species revealed six hub genes governing the maturation and hypertrophy of skeletal muscle: *PPARGC1A, MYOD1, EPAS1, IGF2, CXCR4*, and *APOA1* (Figure 2, Table 11).

The first of them, peroxisome proliferator-activated receptor gamma coactivator 1-alpha (*PPARGC1A*), was identified as the hub gene with the highest degree of connectivity. *PGC-1a* (aka *PPARGC1A*) was reported to be a necessary control factor in skeletal muscle development, adaptation to exercise, the transcriptional control of genes responsible for angiogenesis, fatty acid oxidation, oxidative phosphorylation, mitochondrial biogenesis, and muscle fiber type composition and transition to a slow-twitch muscle type [34,35,36]. *PPARGC1A* is necessary for the proper myogenic differentiation of C2C12 cells, acting as a mediator of mitochondria biogenesis in the emerging myotubes [37]. Moreover, a decrease in *PPARGC1A* expression is accompanied by reactive species generation and mitochondrial damage, which finally results in inefficient differentiation [38]. It also appears to play a protective role against atrophy-linked skeletal muscle deterioration [39]. According to our findings, this gene is involved in the positive regulation of multicellular organismal processes, insulin signaling and the insulin resistance pathway (Table 10), and the processes governing the balance between cell division and differentiation, which determine organ size in multicellular organisms and also influence the balance between skeletal muscle hypertrophy and atrophy [40].

One of the muscle regulatory factors and also a hub gene (Table 11), *MYOD1* is expressed in developing skeletal muscle [20]. The expression of the *MYOD1* gene, a major transcriptional regulator of myogenesis, is detectable in proliferating myoblasts, activated satellite cells, and myocytes [41]. It also promotes the transcription of p21 and myogenin, which allows cells to exit the cell cycle and stop the proliferation of differentiated myocytes [42,43]. *MYOD1* activity with *MYF5* is required for skeletal muscle growth, hypertrophy, and regeneration, which are processes strongly dependent on the activation of satellite cells. The regulation of muscle cell differentiation is the key determinant of the frequency, rate, and extent of skeletal muscle development. It is well known and also confirmed in our analysis that *MYOD1,* one of the other high-grade hub genes, is related to the regulation of muscle cell differentiation, skeletal muscle fiber development, myoblast fusion, myotube cell development, and other processes mentioned in Table 6 and Table 10.

The next identified hub gene is endothelial PAS domain protein 1 (EPAS 1; known as hypoxia inducible factor 2A (HIF2A) (Table 11). HIF2A is a protein that, in an HIF complex, plays an important role in the ability of tissue to adapt to changing oxygen levels [44]. It also regulates the insulin signaling pathway [45], which, as mentioned earlier, has a significant role in metabolism, growth, reproduction, and aging. Our results showed that HIF2A is related to the myoblast differentiation and positive regulation of multicellular organismal processes. The available literature data indicates that HIF2A induces the quiescence and self-renewal of satellite cells, which are hypoxic in the niche, and blocks their myogenic differentiation [46]; however, some studies indicate that HIF2A is preferentially expressed in postdifferentiation myoblasts. Its pharmacological inhibition accelerates muscle regeneration by increasing satellite cells proliferation and differentiation [46,47]. However, some authors observed that the deficiency of HIF1A and HIF2A delayed muscle regeneration by reducing the number of satellite cells [47,48]. HIF2A stimulates the expression of genes encoding antioxidant enzymes, suppressing aberrant ROS accumulation, and its deficiency leads to severely striated muscle damage [44]. It may also modulate glucose metabolism, probably indirectly by PPAR [49]. HIF2A influences skeletal muscle myofiber types and metabolic capacity, encodes an oxidative slow-twitch muscle program in the skeletal muscle, and controls the glycolytic and oxidative metabolism of skeletal muscle fibers together with HIF1A protein [50]. Moreover, HIF2A activation in the skeletal muscle can induce angiogenesis [51]. HIF2A appears to be a potent regulator of processes vital for skeletal muscle development by playing a significant role in controlling, directly or indirectly, processes of proliferation and differentiation, the self-renewal of satellite cells, ROS control, glucose metabolism, and vascularization.

Insulin-like growth factor 2 (*IGF2*) is a hub gene (Table 11). According to our findings, the gene has been associated to the regulation of muscle cell differentiation, positive regulation of the insulin receptor signaling pathway, and regulation of cell population proliferation (Table 6 and Table 10). *IGF2* was previously discovered to regulate the postnatal growth of skeletal muscle and internal organs [52]. *IGF2* is a myogenesis regulator and autocrine factor that initiates myoblast differentiation in vitro and supports muscle fiber formation—the fusion of mononucleotide precursor cells into multinucleated mature cells [53,54,55]. *IGF2* and M*YOD1* are co-regulated during myogenesis, which affects the progression of myogenic differentiation [56]. The anabolic actions of *IGF1* and *IGF2* on skeletal muscle are closely related and are mediated by the same receptor, namely the type 1 IGF receptor (IGF1R; [57]). *IGF2* released by differentiating myoblasts can enhance the transcriptional properties of *MYOD1* [58]. Zhu et al. [59] demonstrated that *IGF2* deficiency in myotubes leads to impaired mitochondria function, manifested by a reduced content of mitochondrial protein, an imbalance of fission/fusion, and impaired biogenesis. The authors proposed that mitochondrial defects may occur through the IGF2-SIRT1-PGC1α(PPARGC1A) pathway [59].

The penultimate hub gene identified as a common gene for *Bos taurus, Ovis aries*, and *Sus scrofa* skeletal muscle maturation and hypertrophy is C-X-C chemokine receptor type 4 (*CXCR4*), a motility-stimulating chemokine receptor [60]. Both the chemokine *SDF1* (also known as *CXCL12*) secreted by myofibers and its primary receptor *CXCR4* promote developmental myogenesis as well as muscle regeneration. *SDF1* attracts *CXCR4*-positive satellite cells, stimulating cell migration, promoting myoblast fusion with existing myofibers, and inducing angiogenesis in regenerating muscles [61,62]. Vasyutina et al. [63] indicated that *CXCR4*-dependent signals also control survival. The *CXCR4/SDF1* axis is involved in cell migration, which is also essential for skeletal muscle repair. Furthermore, as indicated by Ref. [64], the overexpression of *CXCR4/SDF1* can protect cachectic muscle from wasting by increasing the fiber area by 20%. Despite the well-established role of *CXCR4/SDF1* in embryonic muscle development and muscle regeneration, the function of this pathway during adult myogenesis remains to be fully elucidated [65]. Two other hub genes, *EPAS1/HIF2A* and *PPARGC1A*, have already been mentioned as being involved in this process. The interactions of *CXCR4*, *HIF2A*, and *PPAR* are documented in the literature data, proving that the *HIF2A* transcription regulation of *CXCR4* is involved in macrophage migration and chemotaxis [66], as well as the *PPAR*-related downregulation of *CXCR4* gene expression in cancer cells [67].

The last hub gene identified in this study is apolipoprotein A1 (*APOA1*). *APOA1* is the major apolipoprotein component of the high-density lipoprotein (HDL), which is involved in cholesterol transport, the regulation of lipid metabolism and transport, and mitochondria biogenesis and metabolism [68]. It is endogenously expressed by skeletal muscle cells [69]. In the absence of insulin, *APOA1* increases glucose uptake into skeletal muscle cells and increase glucose consumption in skeletal muscles, resulting in improved glucose tolerance [70,71]. Finally, it was also proved that *APOA1* increases glucose disposal in skeletal muscle, thus supporting a role for HDL in reducing insulin resistance [72]. Moreover, the results of Liu et al. [73] suggest that *APOA1* may be useful as a molecular marker of intramuscular fat (IMF) deposition in chickens. Moreover, the APOA1 gene was proposed by Ref. [74] as one of the genes responsible for the IMF content in high marbling samples of longissimus muscle in cattle. In another IMF biomarkers identification study, *APOA1* was clustered to a fat deposition group of genes, with the PPAR signaling and ECM–receptor interaction signaling pathways being shown as the most important [75], which corresponds well with the results obtained in our analysis (Table 6 and Table 10).

Muscle fiber hypertrophy produced by satellite cell activation, proliferation, differentiation, and fusion with existing fibers was the predominant contributor to postnatal muscle growth and meat quality [76]. With the use of previous findings, we have depicted the genes that play a pivotal role in the above-mentioned processes, namely the hub genes shared by the three ruminant species we studied. The aforementioned six hub genes can be used as biomarkers for breeding programs because of their relationship to the maturation and hypertrophy of skeletal muscle, as well as their role in the deposition of intramuscular fat. Their interaction can be the basis for directing the development of skeletal muscles in accordance with the declared demands of breeders or consumers. The summary of the relevance of the identified genes and related processes is presented in Figure 3.

## 5. Conclusions

Our investigation of transcriptome profiles in the muscle tissues of *Bos taurus, Ovis aries*, and *Sus scrofa* showed that common DEGs up-regulated and down-regulated between the described species. Biological pathways were found to be directly related to muscle growth and hypertrophy. The progress of gene annotation studies will allow more precise knowledge representation, which in turn will generate more informative results from data analyses. The results may provide novel insights into targets that can be used for future investigations of underlying molecular mechanisms. Future research should focus on these important genes and pathways in order to definitively pinpoint advantageous biological targets for analyzing skeletal muscle maturation and hypertrophy.

## Figures and Tables

**Figure 1 animals-12-03471-f001:**
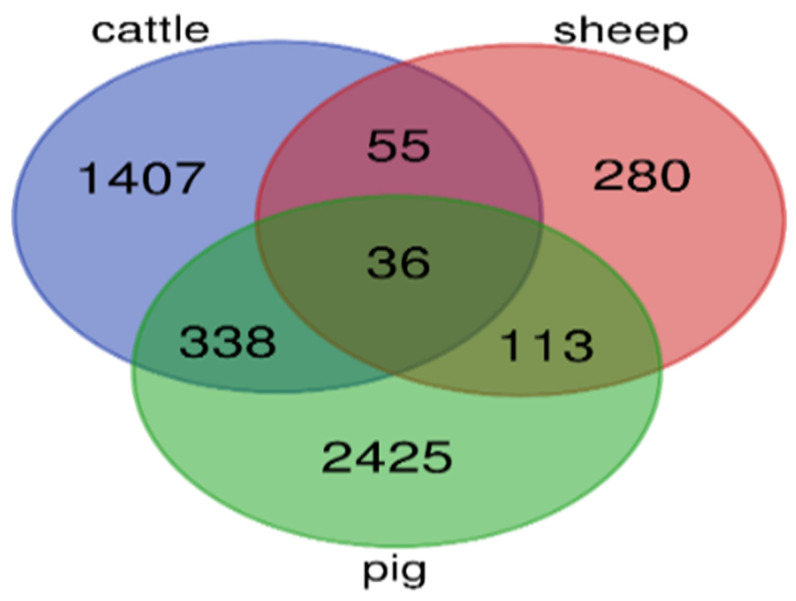
Number of common genes identified between three species using Venn diagram.

**Figure 2 animals-12-03471-f002:**
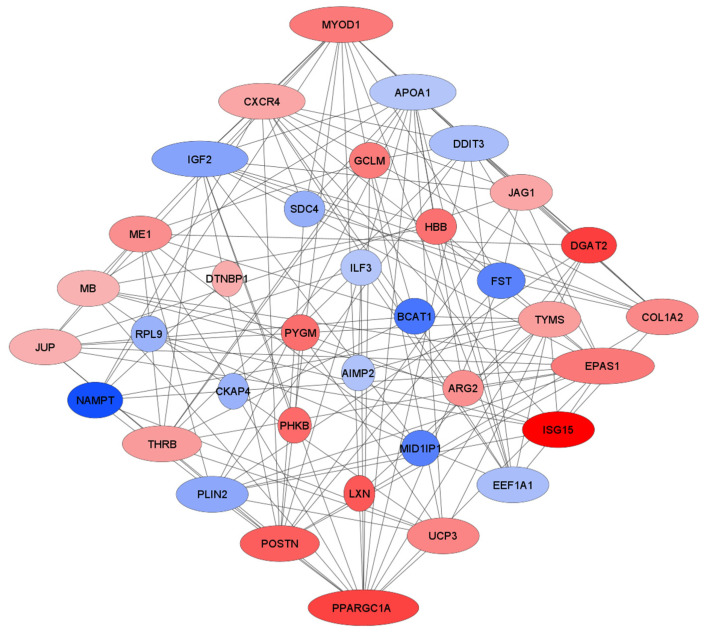
PPI network of differentially expressed genes of skeletal muscle in *Bos taurus, Ovis aries*, and *Sus scrofa*. DEGs—differentially expressed genes; Blue—downregulated DEGs; Red—upregulated DEGs. The node size is proportional to the degree bigger than 12. The bigger size of the node means a higher engagement of protein in the developmental process of skeletal muscle in *Bos taurus, Ovis aries*, and *Sus scrofa*.

**Figure 3 animals-12-03471-f003:**
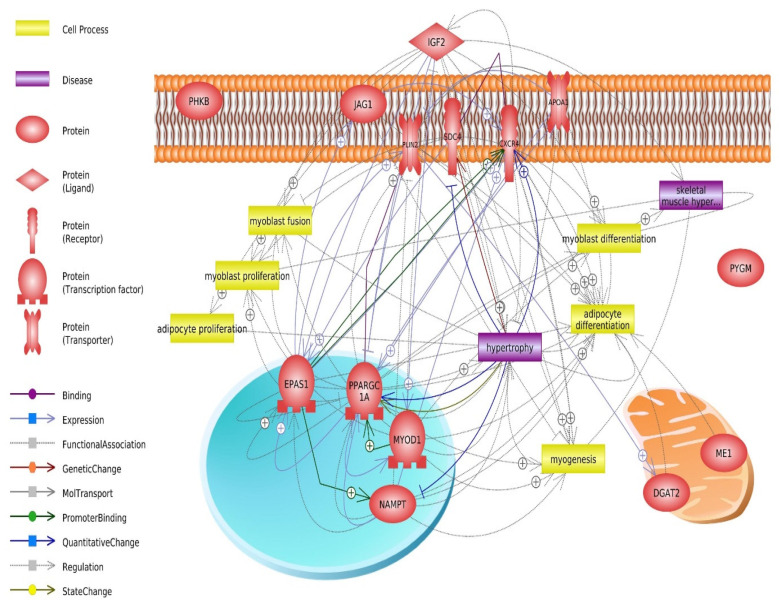
Pathway studio visualization of gene relevance network.

**Table 1 animals-12-03471-t001:** Accession numbers of *Bos taurus, Ovis aries*, and *Sus scrofa* skeletal muscle tissue expression profiles.

Species	Experimental Group	Accession Numbers
*Bos taurus*	280 dpc	GSM1077020 GSM1077030
	3 month	GSM1077021 GSM1077031
*Ovis aries*	135 dpc	GSM578053 GSM578054 GSM578055
	2 month	GSM578050 GSM578051 GSM578052
*Sus scrofa*	91 dpc	GSM944491 GSM944492 GSM944493
	adult	GSM944494 GSM944495 GSM944496

**Table 2 animals-12-03471-t002:** Common genes among *Bos taurus, Ovis aries*, and *Sus scrofa*.

Genes
CXCR4, DTNBP1, GCLM, THRB, PPARGC1A, FST, LXN, TYMS, RPL9, HBB, ARG2, JUP, UCP3, COL1A2, PHKB, ISG15, DGAT2, APOA1, PYGM, PLIN2, MYOD1, NAMPT, POSTN, EPAS1, ME1, CKAP4, MID1IP1, JAG1, SDC4, ILF3, AIMP2, EEF1A1, IGF2, MB, DDIT3, BCAT1

**Table 3 animals-12-03471-t003:** Biological processes associated with skeletal-muscle-maturation- and hypertrophy-related DEGs in *Bos taurus*.

Category	Gene Set	Description	*p*-Value	Count
BP	GO:0030324	Lung development	0.005712	13
BP	GO:0030517	Negative regulation of axon extension	0.008625	5
BP	GO:0048741	Skeletal muscle fiber development	0.011764	7
BP	GO:0003009	Skeletal muscle contraction	0.023003	6
BP	GO:0048146	Positive regulation of fibroblast proliferation	0.02471	9
BP	GO:0048743	Positive regulation of skeletal muscle fiber development	0.02639	4
BP	GO:0045662	Negative regulation of myoblast differentiation	0.029267	6
BP	GO:0045663	Positive regulation of myoblast differentiation	0.029267	6
BP	GO:0007179	Transforming growth factor-beta receptor signaling pathway	0.033284	11

DEGs—differentially expressed genes; BP—biological process; *p*-value ≤ 0.05.

**Table 4 animals-12-03471-t004:** Biological processes associated with skeletal-muscle-maturation- and hypertrophy-related DEGs in *Ovis aries*.

Category	Gene Set	Description	*p*-Value	Count
BP	GO:0042127	Regulation of cell proliferation	0.001109	9
BP	GO:0030307	Positive regulation of cell growth	0.002954	7
BP	GO:0042104	Positive regulation of activated T cell proliferation	0.002995	5
BP	GO:0046697	Decidualization	0.003547	4
BP	GO:0033690	Positive regulation of osteoblast proliferation	0.029267	4
BP	GO:0001938	Positive regulation of endothelial cell proliferation	0.009497	6
BP	GO:0048662	Negative regulation of smooth muscle cell proliferation	0.009759	4
BP	GO:0001558	Regulation of cell growth	0.012546	5
BP	GO:0048741	Skeletal muscle fiber development	0.014303	4
BP	GO:0043568	Positive regulation of insulin-like growth factor receptor signaling pathway	0.020239	3
BP	GO:0048146	Positive regulation of fibroblast proliferation	0.022068	5
BP	GO:0042102	Positive regulation of T cell proliferation	0.032177	5
BP	GO:0035914	Skeletal muscle cell differentiation	0.048064	5

DEGs—differentially expressed genes; BP—biological process; *p*-value ≤ 0.05.

**Table 5 animals-12-03471-t005:** Biological processes associated with skeletal-muscle-maturation- and hypertrophy-related DEGs in *Sus scrofa*.

Category	Gene Set	Description	*p*-Value	Count
BP	GO:0008284	Positive regulation of cell proliferation	0.003942	47
BP	GO:0007507	Heart development	0.007053	26
BP	GO:0048469	Cell maturation	0.009036	10
BP	GO:0000082	G1/S transition of mitotic cell cycle	0.009406	12
BP	GO:0016202	Regulation of striated muscle tissue development	0.016587	4
BP	GO:0048147	Negative regulation of fibroblast proliferation	0.019795	7
BP	GO:0045662	Negative regulation of myoblast differentiation	0.021486	8
BP	GO:0030308	Negative regulation of cell growth	0.022828	20
BP	GO:0008286	Insulin receptor signaling pathway	0.032387	11
BP	GO:0001822	Kidney development	0.042236	11
BP	GO:0009791	Postembryonic development	0.042236	14
BP	GO:0043588	Skin development	0.049737	8

DEGs—differentially expressed genes; BP—biological process; *p*-value ≤ 0.05.

**Table 6 animals-12-03471-t006:** Biological processes associated with skeletal-muscle-maturation- and hypertrophy-related DEGs common for *Bos taurus, Ovis aries*, and *Sus scrofa*.

Category	Description	Genes	*p*-Value
BP	Adipose tissue development	DGAT2, NAMPT	2.06 × 10^−4^
BP	Myoblast differentiation	JAG1, EPAS1	3.26 × 10^−4^
BP	Regulation of muscle cell differentiation	MYOD1, IGF2	0.001805571
BP	Positive regulation of multicellular organismal process	EPAS1, FST, CXCR4, PPARGC1A	0.003315863
BP	Insulin receptor signaling pathway	NAMPT, IGF2	0.007640329
BP	Skin development	JAG1, COL1A2	0.009114935
BP	Positive regulation of skeletal muscle fiber development	MYOD1	0.010752747
BP	Regulation of skeletal muscle fiber development	MYOD1	0.014311969
BP	Negative regulation of muscle cell differentiation	IGF2	0.014311969
BP	Positive regulation of myoblast fusion	MYOD1	0.017858741
BP	Positive regulation of skeletal muscle tissue development	MYOD1	0.017858741
BP	Skeletal muscle cell differentiation	MYOD1	0.019627472
BP	Myotube cell development	MYOD1	0.019627472
BP	Regulation of myoblast fusion	MYOD1	0.021393106
BP	Cardiac cell development	JAG1	0.023155649
BP	Positive regulation of insulin receptor signaling pathway	IGF2	0.023155649
BP	Aorta development	JAG1	0.02491506
BP	Pulmonary valve development	JAG1	0.031922176
BP	Positive regulation of myoblast differentiation	MYOD1	0.031922176
BP	Myotube differentiation	MYOD1	0.03714533
BP	Regulation of cell population proliferation	JAG1, JUP, NAMPT, IGF2	0.047332288
BP	Positive regulation of muscle cell differentiation	MYOD1	0.047509573
BP	Cellular response to epidermal growth factor stimulus	EEF1A1	0.049226369

DEGs—differentially expressed genes; BP—biological process; *p*-value ≤ 0.05.

**Table 7 animals-12-03471-t007:** KEGG pathway analysis of the maturation- and hypertrophy-related DEGs and proportional *p*-values identified in *Bos taurus*.

Databases	Gene Set	Description	*p*-Value	Count
KEGG pathway	bta03008	Ribosome biogenesis in eukaryotes	2.35 × 10^−7^	25
KEGG pathway	bta03320	PPAR signaling pathway	0.002589	16
KEGG pathway	bta04066	HIF-1 signaling pathway	0.004774	19

**Table 8 animals-12-03471-t008:** KEGG pathway analysis of the maturation- and hypertrophy-related DEGs and proportional *p*-values identified in *Ovis aries*.

Databases	Gene Set	Description	*p*-Value	Count
KEGG pathway	oas04512	ECM–receptor interaction	4.35 × 10^−6^	15
KEGG pathway	oas04151	PI3K-Akt signaling pathway	8.79 × 10^−5^	29
KEGG pathway	oas04510	Focal adhesion	9.02 × 10^−5^	21
KEGG pathway	oas04660	T-cell receptor signaling pathway	0.001049	12
KEGG pathway	oas04066	HIF-1 signaling pathway	0.010325	10
KEGG pathway	oas04015	Rap1 signaling pathway	0.020892	15

**Table 9 animals-12-03471-t009:** KEGG pathway analysis of the maturation- and hypertrophy-related DEGs and proportional *p*-values identified in *Sus scrofa*.

Databases	Gene Set	Description	*p*-Value	Count
KEGG pathway	ssc04910	Insulin signaling pathway	8.92 × 10^−5^	40
KEGG pathway	ssc04066	HIF-1 signaling pathway	1.11 × 10^−4^	32
KEGG pathway	ssc05205	Proteoglycans in cancer	1.59 × 10^−4^	52
KEGG pathway	ssc04510	Focal adhesion	2.42 × 10^−4^	52
KEGG pathway	ssc04068	FoxO signaling pathway	0.0012	37
KEGG pathway	ssc04152	AMPK signaling pathway:	0.001827	33
KEGG pathway	ssc04919	Thyroid hormone signaling pathway	0.002029	32
KEGG pathway	ssc04512	ECM–receptor interaction	0.00208	25
KEGG pathway	ssc05410	Hypertrophic cardiomyopathy (HCM)	0.01255	22
KEGG pathway	ssc04151	PI3K-Akt signaling pathway	0.026508	68

**Table 10 animals-12-03471-t010:** KEGG pathways common for *Bos taurus, Ovis aries*, and *Sus scrofa*.

Databases	Description	Gene	*p*-Value
KEGG pathway	PPAR signaling pathway	ME1, APOA1, PLIN2	3.18 × 10^−4^
KEGG pathway	Insulin signaling pathway	PYGM, PHKB, PPARGC1A	0.001903
KEGG pathway	ECM–receptor interaction	COL1A2, SDC4	0.00109943
KEGG pathway	Insulin resistance	PYGM, PPARGC1A	0.016151

**Table 11 animals-12-03471-t011:** Top six hub genes ranked by degree method in *Bos taurus, Ovis aries*, and *Sus scrofa*.

	Betweenness	Degree	Gene Name	Gene
0.625	174.5703	15	Peroxisome proliferator-activated receptor gamma coactivator 1-alpha	PPARGC1A
0.59322	76.05031	14	Myoblast determination protein 1	MYOD1
0.59322	73.11194	14	Endothelial PAS domain protein 1	EPAS1
0.546875	99.93942	13	Insulin-like growth factor 2	IGF2
0.555556	49.89188	12	C-X-C chemokine receptor type 4	CXCR4
0.564516	65.64227	12	Apolipoprotein AI	APOA1

## Data Availability

Not applicable.

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
