# Peer review of "Identification of Key Genes and Biological Pathways Associated with Skeletal Muscle Maturation and Hypertrophy in Bos taurus, Ovis aries, and Sus scrofa"

_animals, 2022, doi:10.3390/ani12243471_

Round 1

Reviewer 1 Report (Previous Reviewer 4)

The authors have addressed all my concerns. Therefore, it is recommended to accept this paper for publication in Animals.

I have one suggestion, change the title to "Identification of key genes and biological pathways......".

Author Response

Responses to Reviewers’ comments

Manuscript title: Identification of key genes and biological pathways associated with skeletal muscle maturation and hypertrophy in Bos taurus, Ovis aries, and Sus scrofa

Thank you for giving us the opportunity to submit a revised draft of manuscript. The authors appreciate the time and effort that you and the reviewers have dedicated to providing your valuable feedback and  for their insightful comments on the manuscript. We have been able to incorporate changes to reflect most of the suggestions provided by the reviewers.

Reviewer #1

The authors would like to thank the Reviewer for constructive comments that contributed to improving the quality of the manuscript.

R1 comment: I have one suggestion, change the title to "Identification of key genes and biological pathways......".

According to Reviever’s suggestion the title was changed to “Identification of key genes and biological pathways associated with skeletal muscle maturation and hypertrophy in Bos taurus, Ovis aries, and Sus scrofa

Reviewer #3

The authors would like to thank the Reviewer for constructive comments that contributed to improving the quality of the manuscript.

R3 comment:  1. Highlighted text - I couldn’t understand why some of the text and references were highlighted in this final draft. Can you please elaborate that?

The manuscript was resubmitted to the journal as previously suggested by the Editor and eviewers. Changes from version 1 are marked in green in v2.

R3 comment: 2. Spacing changes - Paragrpah-1 and rest of the introductory paragraphs have different spacings as apparent from the text. Why is it that way, can you please explain?

We apologize for overlooking the editorial error in the form of different spacing between lines. Most likely, the text at this point was formatted in the same way as the preceding abstract.

R3 comment: 3. Line 52-55 - Meat also includes visceral organs albeit termed as visceral meat. I suggest mentioning it here as we are talking about meat and not muscle meat only.

We changed the sentences according to Reviewer suggestion:
“Meat represents a major source of animal protein in the human diet (both skeletal muscle and visceral meat). Muscle meat is mainly composed of skeletal muscle fibers and various amounts of connective tissue, adipose tissue, as well as small amounts of epithelial and nervous tissue which play a role in its quantitative and qualitative properties [1].”

R3 comment: 4. Paragraph-3 - I would suggest adding the short definitions of the hyperplasia and hypertrophy in the previous paragraph (much like in the discussion Line 232-234). I don’t think a full paragraph of only definitions in warranted here.

We added short description of hyperplasia and hypertrophy in introduction, as suggested by the Reviewer.

“Livestock muscle is developed in two main stages: the prenatal phase – proliferation (hyperplasia - an increase in number of cells) as well as myotube formation, and the postnatal phase – muscle growth by hypertrophy (an increase in size of muscle fibers) [6].”

R3 comment: 5. Can you please elaborate why you chose these species and not others? Was there a specific reason behind it?

For the study, we selected three species of livestock animals, ruminants, which have an established worldwide position as a source of meat.

https://ourworldindata.org/grapher/global-meat-production-by-livestock-type

Comparison of gene expression among muscles of farm animals such as poultry, pigs, horses and ruminants may provide additional knowledge about more conservative mechanisms of regulation of skeletal muscle development. It is possible that in the future we will attempt to compare these species.

R3 comment: 6. Introduction is precise but I think starting the paper with a much gross idea as meat consumption may not be appropriate for a paper of this nature and analysis. I would recommend modifying your opening pitch.

According to Reviever’s suggestion the first paragraph of the introduction was changed

R3 comment: 7. Will this type of research involve explicit ethics approval? If not, please mention in the methodology.

It was added at the end of manuscript: “Institutional Review Board Statement: This research was carried out on publicly available data deposited in the NCBI GEO database. Therefore, the approval of the ethics committee was not required.”

  1. Discussion is well written, but it is way too long and can make the reader a bit overwhelmed at times. Can you please try to decrease the length if possible?

We are aware of the length of the discussion. In the discussion, we briefly described the most important biological processes and signaling pathways. We devoted a little more attention to the 6 hub genes. It will be difficult to shorten the discussion due to the extensive involvement of the identified genes in the processes described in the discussion. We have tried to present them as concisely as possible, and at the same time indicate their broad contribution to the development of skeletal muscles and the possibilities of their mutual interaction. We will be grateful if you keep the discussion in its current form.

Best Regards

Corresponding Authors

Reviewer 2 Report (Previous Reviewer 2)

Dear authors

the new form for me is OK, you have followed my suggestions.

regards

Cinzia

Author Response

Responses to Reviewers’ comments

Manuscript title: Identification of key genes and biological pathways associated with skeletal muscle maturation and hypertrophy in Bos taurus, Ovis aries, and Sus scrofa

Thank you for giving us the opportunity to submit a revised draft of manuscript. The authors appreciate the time and effort that you and the reviewers have dedicated to providing your valuable feedback and  for their insightful comments on the manuscript. We have been able to incorporate changes to reflect most of the suggestions provided by the reviewers.

Reviewer #1

The authors would like to thank the Reviewer for constructive comments that contributed to improving the quality of the manuscript.

R1 comment: I have one suggestion, change the title to "Identification of key genes and biological pathways......".

According to Reviever’s suggestion the title was changed to “Identification of key genes and biological pathways associated with skeletal muscle maturation and hypertrophy in Bos taurus, Ovis aries, and Sus scrofa

Reviewer #3

The authors would like to thank the Reviewer for constructive comments that contributed to improving the quality of the manuscript.

R3 comment:  1. Highlighted text - I couldn’t understand why some of the text and references were highlighted in this final draft. Can you please elaborate that?

The manuscript was resubmitted to the journal as previously suggested by the Editor and eviewers. Changes from version 1 are marked in green in v2.

R3 comment: 2. Spacing changes - Paragrpah-1 and rest of the introductory paragraphs have different spacings as apparent from the text. Why is it that way, can you please explain?

We apologize for overlooking the editorial error in the form of different spacing between lines. Most likely, the text at this point was formatted in the same way as the preceding abstract.

R3 comment: 3. Line 52-55 - Meat also includes visceral organs albeit termed as visceral meat. I suggest mentioning it here as we are talking about meat and not muscle meat only.

We changed the sentences according to Reviewer suggestion:
“Meat represents a major source of animal protein in the human diet (both skeletal muscle and visceral meat). Muscle meat is mainly composed of skeletal muscle fibers and various amounts of connective tissue, adipose tissue, as well as small amounts of epithelial and nervous tissue which play a role in its quantitative and qualitative properties [1].”

R3 comment: 4. Paragraph-3 - I would suggest adding the short definitions of the hyperplasia and hypertrophy in the previous paragraph (much like in the discussion Line 232-234). I don’t think a full paragraph of only definitions in warranted here.

We added short description of hyperplasia and hypertrophy in introduction, as suggested by the Reviewer.

“Livestock muscle is developed in two main stages: the prenatal phase – proliferation (hyperplasia - an increase in number of cells) as well as myotube formation, and the postnatal phase – muscle growth by hypertrophy (an increase in size of muscle fibers) [6].”

R3 comment: 5. Can you please elaborate why you chose these species and not others? Was there a specific reason behind it?

For the study, we selected three species of livestock animals, ruminants, which have an established worldwide position as a source of meat.

https://ourworldindata.org/grapher/global-meat-production-by-livestock-type

Comparison of gene expression among muscles of farm animals such as poultry, pigs, horses and ruminants may provide additional knowledge about more conservative mechanisms of regulation of skeletal muscle development. It is possible that in the future we will attempt to compare these species.

R3 comment: 6. Introduction is precise but I think starting the paper with a much gross idea as meat consumption may not be appropriate for a paper of this nature and analysis. I would recommend modifying your opening pitch.

According to Reviever’s suggestion the first paragraph of the introduction was changed

R3 comment: 7. Will this type of research involve explicit ethics approval? If not, please mention in the methodology.

It was added at the end of manuscript: “Institutional Review Board Statement: This research was carried out on publicly available data deposited in the NCBI GEO database. Therefore, the approval of the ethics committee was not required.”

  1. Discussion is well written, but it is way too long and can make the reader a bit overwhelmed at times. Can you please try to decrease the length if possible?

We are aware of the length of the discussion. In the discussion, we briefly described the most important biological processes and signaling pathways. We devoted a little more attention to the 6 hub genes. It will be difficult to shorten the discussion due to the extensive involvement of the identified genes in the processes described in the discussion. We have tried to present them as concisely as possible, and at the same time indicate their broad contribution to the development of skeletal muscles and the possibilities of their mutual interaction. We will be grateful if you keep the discussion in its current form.

Best Regards

Corresponding Authors

Reviewer 3 Report (Previous Reviewer 1)

Author Response

Responses to Reviewers’ comments

Manuscript title: Identification of key genes and biological pathways associated with skeletal muscle maturation and hypertrophy in Bos taurus, Ovis aries, and Sus scrofa

Thank you for giving us the opportunity to submit a revised draft of manuscript. The authors appreciate the time and effort that you and the reviewers have dedicated to providing your valuable feedback and  for their insightful comments on the manuscript. We have been able to incorporate changes to reflect most of the suggestions provided by the reviewers.

Reviewer #1

The authors would like to thank the Reviewer for constructive comments that contributed to improving the quality of the manuscript.

R1 comment: I have one suggestion, change the title to "Identification of key genes and biological pathways......".

According to Reviever’s suggestion the title was changed to “Identification of key genes and biological pathways associated with skeletal muscle maturation and hypertrophy in Bos taurus, Ovis aries, and Sus scrofa

Reviewer #3

The authors would like to thank the Reviewer for constructive comments that contributed to improving the quality of the manuscript.

R3 comment:  1. Highlighted text - I couldn’t understand why some of the text and references were highlighted in this final draft. Can you please elaborate that?

The manuscript was resubmitted to the journal as previously suggested by the Editor and eviewers. Changes from version 1 are marked in green in v2.

R3 comment: 2. Spacing changes - Paragrpah-1 and rest of the introductory paragraphs have different spacings as apparent from the text. Why is it that way, can you please explain?

We apologize for overlooking the editorial error in the form of different spacing between lines. Most likely, the text at this point was formatted in the same way as the preceding abstract.

R3 comment: 3. Line 52-55 - Meat also includes visceral organs albeit termed as visceral meat. I suggest mentioning it here as we are talking about meat and not muscle meat only.

We changed the sentences according to Reviewer suggestion:
“Meat represents a major source of animal protein in the human diet (both skeletal muscle and visceral meat). Muscle meat is mainly composed of skeletal muscle fibers and various amounts of connective tissue, adipose tissue, as well as small amounts of epithelial and nervous tissue which play a role in its quantitative and qualitative properties [1].”

R3 comment: 4. Paragraph-3 - I would suggest adding the short definitions of the hyperplasia and hypertrophy in the previous paragraph (much like in the discussion Line 232-234). I don’t think a full paragraph of only definitions in warranted here.

We added short description of hyperplasia and hypertrophy in introduction, as suggested by the Reviewer.

“Livestock muscle is developed in two main stages: the prenatal phase – proliferation (hyperplasia - an increase in number of cells) as well as myotube formation, and the postnatal phase – muscle growth by hypertrophy (an increase in size of muscle fibers) [6].”

R3 comment: 5. Can you please elaborate why you chose these species and not others? Was there a specific reason behind it?

For the study, we selected three species of livestock animals, ruminants, which have an established worldwide position as a source of meat.

https://ourworldindata.org/grapher/global-meat-production-by-livestock-type

Comparison of gene expression among muscles of farm animals such as poultry, pigs, horses and ruminants may provide additional knowledge about more conservative mechanisms of regulation of skeletal muscle development. It is possible that in the future we will attempt to compare these species.

R3 comment: 6. Introduction is precise but I think starting the paper with a much gross idea as meat consumption may not be appropriate for a paper of this nature and analysis. I would recommend modifying your opening pitch.

According to Reviever’s suggestion the first paragraph of the introduction was changed

R3 comment: 7. Will this type of research involve explicit ethics approval? If not, please mention in the methodology.

It was added at the end of manuscript: “Institutional Review Board Statement: This research was carried out on publicly available data deposited in the NCBI GEO database. Therefore, the approval of the ethics committee was not required.”

  1. Discussion is well written, but it is way too long and can make the reader a bit overwhelmed at times. Can you please try to decrease the length if possible?

We are aware of the length of the discussion. In the discussion, we briefly described the most important biological processes and signaling pathways. We devoted a little more attention to the 6 hub genes. It will be difficult to shorten the discussion due to the extensive involvement of the identified genes in the processes described in the discussion. We have tried to present them as concisely as possible, and at the same time indicate their broad contribution to the development of skeletal muscles and the possibilities of their mutual interaction. We will be grateful if you keep the discussion in its current form.

Best Regards

Corresponding Authors

This manuscript is a resubmission of an earlier submission. The following is a list of the peer review reports and author responses from that submission.

Round 1

Reviewer 2 Report

Dear authors

I really want to compliment you on the following things: originality for comparing the three species, clarity in outlining the current state of the art, materials and methods, and results and discussion.

Since I am a researcher who puts a lot of work into producing the raw data that is deposited in databases like GEO, I have high moral standards for anyone who want to use it at their discretion. However, this is the open access data policy.

I spent a lot of time determining whether each reference was pertinent to the sentence's assertion, so a word of advice for the future: double-check all numbering and references.

The growth of the muscle and the genes involved are two areas with vast bibliographies, which is another reason why it is preferable to include reviews when discussing issues in general.

Finally, I'd like to know why the paper makes no mention of the hub gene's connection to the GDF8 gene, which generates the myostatin protein.

It has been established that in the three species indicated, GDF8 gene mutations are linked to the development of muscle hypertropia.

For what reason, in my opinion, it is preferable that you include a bibliography on the GDF8 gene and myostatin, as people who, like me, have worked on muscular hypertrophy in the past, would assume that there is a brief statement about it.

You can find following my punctual corrections:

Line 51: substitute reference 48 with many more recent publications, that are available

Lines 51-52: this phrase has to rewrite. I suggest: The fundamental mechanism underlying livestock growth is the accumulation of muscle and adipose tissue.

Line 56: : substitute reference 33 becuase it’s inappropriate

Line 57: substitute reference 47 becuase it’s inappropriate

Line 60: substitute reference 52 becuase it’s inappropriate. I believe there is an error; perhaps you could add reference 51.

Line 63: substitute reference 55 becuase it’s inappropriate.

Line 77: substitute reference 57 becuase it’s inappropriate. There are many more recent publications

Line 78: substitute reference 41 becuase it’s inappropriate. There are many more recent publications

Line 112: substitute reference 56 with 55

Line 122: substitute reference 60 with 59

Line 122: What does mean phrase “The PPI ……..experiments”?

Line 122: substitute reference 19 with 18

Line 124: substitute reference 53 with 52

Line 150: phrase “The most significant ……contraction “. It’s no true because GOs linked to muscle development have  p-values that are less significant than the firt two.

I suggest to write: Among BPs more significant there are processes connected with myogenesis……

Line 183-185: phrase “The most important pathways…….Table 7” I suggest writing as: The most significant pathways are listed in Table 7

Line 189-191: phrase “The results revealed that …..Table 8” I suggest to write as: The most important pathways are presented in Table 8

Line 199-201: phrase “The list of important …….Table 10” I suggest writing: The list of most significant pathways are listed in Table 10.

Line 203-204: phrase “A PPI network was … software” I suggest writing: The PPI network was created using the 36 common DEGs by String database and Cytoscape software.

Line 222: substitute reference 12 with 11

Line 263: reference 27 is inappropriate, Is reference 26 correct?

Line 263: in Table 6, I found only MYOD1 as MRFs

Line 265: substitute reference 4 becuase it’s inappropriate

Line 267: substitute reference 37 becuase it’s inappropriate

Line 268: substitute reference 14, 74 becuase they are inappropriate

Line 269: substitute reference 4 becuase it’s inappropriate

Line 273: delete phrase: “They are described later in the discussion”. This section is discussion yet

Line 275: Are you sure of reference 49?

Line 278: Are you sure of reference 54?

Line 280: substitute reference 63 with 62

Line 283: substitute reference 22 with 21

Line 296: substitute reference 34 with 33

Line 300: exact references as 38,44,45,49

Line 301: Are you sure of reference 71?

Line 304: Are you sure of reference 30?

Line 320: Are you sure of reference 26? I suggest to read the following publication: Journal of Neuromuscular Diseases 8 (2021) 169–183 DOI 10.3233/JND-200568

Line 322: substitute reference 58 with 57

Line 324: substitute reference 43 with 42

Line 325: substitute reference 25 with 24

Line 330: Are you sure of references? I suggest to introduce a publication as a review on cell division and differentiation.

Line 331-343: in all paragraph you should control references, because are all inappropriates. I suggest to introduce only one publication as a review on MYOD1 function.

Line 346: substitute reference 31 with 30

Line 347: substitute reference 64 with 63

Line 352: substitute reference 67 with 66

Line 356: substitute reference 32 with 31

Line 359: delete all references, and introduce a publication about link between HIF2A and PPAR

Line 359: delete  reference 5 it’s inappropriate

Line 363: substitute reference 38 with 37

Line 371: : substitute reference 70 with 69

Line 373: delete reference 2, subsitute referneces 18 e 72 with 17 and 71, respectively

Line 375: substitute reference 8 with 7

Line 376: subsitute reference 24 with 23

Line 377: subsitute reference 65 with 64

Line 378-379: phrase “Moreover …..formation. I suggest delete this phrase becuase You've already mentioned that concept.

Line 379: subsitute reference 76 with 75

Line 383: delete references 76

Line 386: substitute reference 6 with 5

Line 390: substitute reference 9 with 8, delete 69 because it’s inappropriate, subsitute 62 with 61

Line 393: subsitute reference 35 with 34

Line 400: delete reference 31 it’s inappropriate

Line 401: delete reference 47 it’s inappropriate

Line 405: delete reference 29 and introduce a reference that explain APOA1 function in general. Subsitute refernce 73 with 72

Line 408: subsitute reference 15 and 20 with 14 and 19 respectively

Line 409: subsitute reference 61 with 60

Line 410: substitute reference 29 with 28

Line 411:substitute reference 11 with 10

Line 415: substitute reference 28 with 27

Line 419: subsitute refrence 75 with 74

Reviewer 3 Report

Revision of manuscript entitled “Identification of biomarkers and biological pathways associated with skeletal muscle maturation and hypertrophy in cattle, sheep, and pigs”.

 The authors examined gene expression profiles of cattle, pig and sheep muscles, and performed extensive analyses, with the aim of identyfiing hub genes involved in regulation of skeletal muscle development.

The manuscript is well written and interesting to the field.

Reviewer 4 Report

The present study identified DEGs and biological pathways related to muscle maturation and hypertrophy in cattle, sheep, and pigs based on datasets from the GEO database. The findings contribute to a better understanding of the mechanisms underlying livestock muscle maturation and hypertrophy. However, I think that the present manuscript, at last the present version, can not accept for publication in Animals. The English writing needs to be substantially improved.  The references cited in the text are confusing, especially the reference order. For example, the first reference cited in the text is numbered 48, and the second is numbered 21. Many references cited in the text are inconsistent with the Reference section. The authors should carefully improve their writing and prepare the manuscript according to the Instructions for Authors.

1. Line 6, please correct the Affiliation information.

2. The Simple Summary must be improved. A Simple Summary should contain a clear statement of the problem addressed, the aims and objectives, pertinent results, and conclusions from the study.

3. Line 22. P-value < 0.05.

4. Lines 64, 108, 128, please add a reference for sheep.

5. The authors used DAVID (version 6.8) to perform GO and pathway enrichment analysis. I suspect the performance of DAVID in functional enrichment analysis. Results obtained by using DAVID are controversial.

The DAVID version 6.8 has been retired. The authors should use the latest version of DAVID to perform GO and pathway enrichment analysis.

I suggest using the R package clusterProfiler as an alternative.

6. Line 125, did the authors perform differential analysis in the network analysis section?

7. Line 138, why the authors analyzed the common DEGs among cattle, sheep, and pig? Maybe the genes involved in skeletal muscle development are different among them.